# Marine n-3 fatty acid consumption in a Norwegian renal transplant cohort: Comparison of a food frequency questionnaire with plasma phospholipid marine n-3 levels

Joe Chan[1,2]*, My Svensson[1,2], Trond Jenssen[2,3], Erik B. Schmidt[4], Ivar A. Eide[1,3]

1 Department of Renal Medicine, Akershus University Hospital, Lørenskog, Norway, 2 Institute of Clinical Medicine, Faculty of Medicine, University of Oslo, Oslo, Norway, 3 Department of Transplantation Medicine, Oslo University Hospital, Rikshospitalet, Oslo, Norway, 4 Department of Clinical Medicine, Aalborg University Hospital, Aalborg, Denmark

* joe.chan@ahus.no

**Data Availability Statement:** All relevant data are within the paper and its Supporting Information files.

## Abstract

### Background

High levels of plasma marine n-3 fatty acids (n-3FAs) are associated with improved patient and graft survival in renal transplant recipients (RTRs). The aim of this study was to evaluate the utility of a new food frequency questionnaire (FFQ) to estimate marine n-3FA consumption in future epidemiological research.

### Methods

We developed an FFQ with a simple design of 10 questions to assess intake of marine sources of n-3FAs. RTRs included in the recent ORENTRA trial (n = 132) completed the study FFQ at the baseline visit eight weeks after engraftment and at the end of study visit one year post-transplant. We measured the reference biomarker plasma phospholipid (PL) marine n-3FA levels by gas chromatography at the same time points to evaluate association and degree of agreement between FFQ based marine n-3FA consumption estimates and the biomarker.

### Results

The median plasma PL marine n-3FA level was 6.0 weight percentage (wt)% (interquartile range [IQR] 4.7 to 7.3) at baseline and 6.3 wt% (IQR 4.8 to 7.4) at end of study. Median FFQ based marine n-3FA consumption estimates were 22.8 g/month (IQR 13.0 to 34.0) at baseline and 20.3 g/month (IQR 14.5 to 32.3) at end of study. FFQ based marine n-3FA consumption estimates showed a moderate correlation with plasma PL marine n-3FA levels at baseline (Spearman's correlation coefficient $r_s$ = 0.43, p<0.001) and a stronger correlation at end of study ($r_s$ = 0.62, p<0.001). Bland Altman plots showed a reasonable degree of agreement between the two methods at both time points.

**Funding:** The author(s) received no specific funding for this work.

**Competing interests:** The authors have declared that no competing interests exist.

## Conclusions

Marine n-3FA consumption estimates based on the FFQ showed a moderate correlation with the reference biomarker plasma PL marine n-3FA levels. The FFQ might be useful in epidemiological studies where resources are limited.

## Introduction

Marine n-3 fatty acid (n-3FA) consumption may benefit cardiovascular health and renal function following renal transplantation [1, 2]. Previous clinical trials in renal transplant recipients (RTRs) report lower triglyceride levels, higher high-density lipoprotein cholesterol levels and lower diastolic blood pressure after marine n-3FA supplementation [1]. A large cohort study in Norwegian RTRs showed that high plasma phospholipid (PL) n-3FA levels were associated with improved patient and graft survival [3, 4]. Antifibrotic and renoprotective effects of long-term high-dose marine n-3FA supplementation have also been shown for other cardiovascular high-risk populations like myocardial infarction survivors [5, 6]. The recent "Omega-3 fatty acids in Renal Transplantation (ORENTRA)" trial performed in Norwegian RTRs found lower levels of inflammatory biomarkers, less development of renal graft fibrosis and improvement of endothelial function, as well as reduced triglyceride levels after 44 weeks of high-dose n-3FA supplementation [2].

Observational studies and randomized clinical trials (RCTs) studying the influence of marine n-3FA intake on cardiovascular health report conflicting results [7–14]. But a recent meta-analysis, which included three recent large RCTs [15–17], concluded that marine n-3FA supplementation was associated with a lower risk of cardiovascular events and death [18]. In renal transplantation, further studies are warranted to evaluate to what extent marine n-3FA consumption may improve patient and graft survival.

The major marine n-3FAs eicosapentaenoic acid (EPA) and docosahexaenoic acid (DHA) are found in fish and other seafood. Plasma PL levels of EPA and DHA can be measured by fatty acid analysis and are considered valid and reliable measures of marine n-3FA consumption [19]. However, fatty acid analysis is more expensive and time-consuming to apply than a food frequency questionnaire (FFQ). Hence, replacing fatty acid analyses with an FFQ focused on marine n-3FA consumption seems attractive in epidemiological research, provided that the FFQ values show a high degree of agreement and association with the reference biomarker.

The main objective of this study was therefore to evaluate the utility of a new FFQ focused on marine n-3FA consumption, using plasma PL marine n-3FA level as the reference biomarker.

## Materials and methods

### Study participants and design

The study cohort consisted of 132 adult Norwegian RTRs included in the ORENTRA trial [2], who were randomized to receive daily supplementation of either 2.6 g of marine n-3FAs (EPA plus DHA) or 3 g of extra virgin olive oil (control oil) for 44 weeks. All patients gave written informed consent for participation in the trial, which also comprised the study FFQ and fatty acid analysis. The study was approved by the Regional Committees for Medical and Health Research Ethics in Norway and was performed in accordance with the Declaration of Helsinki (Clinical.Trials.gov identifier NCT01744067). FFQ and fatty acid analysis were performed

eight weeks post-engraftment (baseline visit) and one year after transplantation (end of study visit). Patients were treated with standard triple maintenance immunosuppressive regimen consisting of prednisolone, mycophenolate and tacrolimus. Blood samples were drawn in a fasting state in the morning at the baseline and end of study visits. Gas chromatography was used to determine individual fatty acid levels in plasma PLs, quantified as weight percentage (wt%) of total plasma PL fatty acids. We defined marine n-3FA level as the sum of EPA and DHA. The study was performed at Oslo University Hospital during 2012–2015. Details regarding recruitment of patients, fatty acid analysis and the study FFQ are provided in the S1 File.

For the ORENTRA trial, we developed a specific FFQ with a simple design of 10 multiple-choice questions (Fig 1), focusing on food items containing marine sources of n-3FA that are typically found in a Nordic diet [20].

We used three different approaches to estimate marine n-3FA consumption at baseline and end of study based on the FFQ recordings:

1. Marine n-3FA consumption estimates, calculated by combining data from the FFQ with known content of EPA and DHA in fish and other seafoods [21], assuming a standard portion size for a Norwegian population (S1 File, S2 Fig).

2. Marine n-3FA consumption estimates calculated as in approach 1 using only data on fatty fish intake for lunch and dinner (S3 Fig).

3. Number of fish servings per month (S4 Fig).

## Statistical analysis

We used correlation analysis (Spearman's correlation coefficient [$r_s$]) and multivariate regression analysis (data presented as standardized regression coefficients [Std. β-coeff.]) to study associations between FFQ based marine n-3FA consumption estimates and plasma PL marine n-3FA levels. Data obtained by the reference biomarker and the study FFQ were standardized using z-statistics to produce data for both methods on the same scale. This allowed for a more meaningful visual presentation (scatter plots) and made it possible to analyze degree of agreement using Bland Altman plots and one-sample t-test. Since the study drug used in the ORENTRA trial was high-dose marine n-3FA supplementation, we excluded patients in the intervention group when performing statistical analysis of data from the end of study visit. Two patients belonging to the control group did not meet at the end of study visit (n = 66 at baseline, n = 64 at end of study visit). Patient characteristics at baseline grouped according to plasma PL marine n-3FA tertiles were evaluated with analysis of variance for continuous data and Mantel-Haenszel linear-by-linear-trend for categorical data. A two-sided p-value of < 0.05 was considered statistically significant. We used SPSS® version 25.0 (IBM, New York, NY, US) for statistical analyses.

## Results

Patient characteristics for the study cohort have previously been published in detail [2]. Selected variables, grouped according to plasma PL n-3FA tertiles at baseline eight weeks post-transplant, are presented in Table 1. Patients in the upper tertile were older and less often current smokers. Supplementation with cod liver oil was used by 28% of patients in the upper tertile compared with 9% in the lower. Median plasma PL n-3FA levels were 6.0 wt% (interquartile range [IQR] 4.7 to 7.3, n = 132) at the baseline visit and 6.3 wt% (IQR 4.8 to 7.4, n = 64) at the end of study visit. Median FFQ based marine n-3FA consumption estimates were 22.8 g/month (IQR 13.0 to 34.0, n = 132) at baseline and 20.3 g/month (IQR 14.5 to 32.3,

# Food frequency questionnaire for the ORENTRA study

Randomization number

Patient initials

Date (day/month/year)      |   |   |   | 2 | 0 |   |   |

In conjunction with your participation in the ORENTRA study, we ask you to answer the 10 questions below about your eating habits in regards to omega-3 fatty acid intake. Please choose only one answer per question. If you are uncertain about what you eat in a typical month, base your answers on your food intake in the past month. Please answer as honestly as possible. Participation is voluntary.

| During a typical month, how often do you eat these food items? | Never | Seldom | 1-2 times per month | 3-4 times per month | 2-3 times per week | >3 times per week |
|---|---|---|---|---|---|---|
| Herring for dinner? | ☐ | ☐ | ☐ | ☐ | ☐ | ☐ |
| Fatty fish like salmon, trout, sardine or mackerel for dinner? | ☐ | ☐ | ☐ | ☐ | ☐ | ☐ |
| Tuna, halibut, plaice or flounder for dinner? | ☐ | ☐ | ☐ | ☐ | ☐ | ☐ |
| Lean fish like redfish, catfish, pollack and cod for dinner? | ☐ | ☐ | ☐ | ☐ | ☐ | ☐ |
| Food made of fish paste, fish gratin and breaded fish for dinner? | ☐ | ☐ | ☐ | ☐ | ☐ | ☐ |
| Other seafood like crab, mussels, shrimps and lobster for dinner? | ☐ | ☐ | ☐ | ☐ | ☐ | ☐ |
| Fatty fish (herring, salmon, sardine and anchovy) as bread spread? | ☐ | ☐ | ☐ | ☐ | ☐ | ☐ |
| Lean fish (cod, tuna and food made of fish paste) as bread spread? | ☐ | ☐ | ☐ | ☐ | ☐ | ☐ |
| Other seafood (crab, shrimps and crayfish tails) as bread spread? | ☐ | ☐ | ☐ | ☐ | ☐ | ☐ |
| Cod liver oil or omega-3 in liquid or capsulated form? | ☐ | ☐ | ☐ | ☐ | ☐ | ☐ |

**Fig 1. Study food frequency questionnaire focusing on food items containing marine n-3 polyunsaturated fatty acids (English version).** The study subjects responded to the question "During a typical month, how often do you eat these food items?" using one of six response alternatives for each of the ten food item categories.

**Table 1. Patient characteristics at baseline eight weeks after renal transplantation according to plasma phospholipid marine n-3 fatty acid tertiles.**

| Variables | All patients | Plasma PL marine n-3FA level, wt% | | | p (trend) |
|---|---|---|---|---|---|
| | | ≤5.1 | 5.2–6.9 | ≥7.0 | |
| Number of patients | 132 | 44 | 44 | 44 | |
| FFQ based marine fatty acid consumption estimate, g/month | 26.0 (16.6) | 19.8 (15.0) | 25.4 (17.5) | 33.1 (14.7) | 0.001 |
| Number of servings of fish / month | 19.9 (15.6) | 13.3 (13.8) | 16.2 (13.4) | 26.9 (16.8) | 0.12 |
| Marine n-3FA supplements, % | 14.5 | 9.1 | 6.8 | 27.9 | 0.01 |
| Recipient age, years | 53.4 (13.8) | 45.7 (12.6) | 55.6 (13.6) | 59.1 (12.0) | <0.001 |
| Recipient gender (Female), % | 25.8 | 27.3 | 25.0 | 25.6 | 0.86 |
| Ethnicity, White, % | 92.4 | 93.2 | 86.0 | 97.7 | 0.39 |
| Body mass index, kg/m² | 26.0 (3.9) | 25.2 (4.0) | 25.9 (3.8) | 26.8 (3.7) | 0.16 |
| Educational level, % | | | | | |
| >3 years at University | 29.8 | 27.3 | 27.3 | 34.9 | |
| 1–3 years at University | 7.6 | 6.8 | 6.8 | 9.3 | |
| Secondary school | 35.1 | 40.9 | 36.4 | 27.9 | |
| Primary school | 27.5 | 25.0 | 29.5 | 27.9 | 0.56 |
| Physical exercise, % | | | | | |
| High intensity ≥ twice per week | 42.1 | 51.2 | 36.4 | 39.0 | |
| High intensity once per week | 9.5 | 2.4 | 11.4 | 14.6 | |
| Low intensity ≥ twice per week | 34.9 | 36.6 | 36.4 | 31.7 | |
| Low intensity once per week | 7.1 | 9.8 | 9.1 | 2.4 | |
| None | 6.3 | 0.0 | 6.8 | 12.2 | 0.29 |
| Smoking habits, % | | | | | |
| Daily smoker | 12.7 | 19.5 | 11.4 | 7.3 | |
| Non-daily smoker | 3.2 | 7.3 | 0.0 | 2.4 | |
| Former heavy smoker | 7.1 | 7.3 | 4.5 | 9.8 | |
| Former light smoker | 35.7 | 34.1 | 43.2 | 29.3 | |
| Life-long non-smoker | 41.3 | 31.7 | 40.9 | 51.2 | 0.03 |

Patient characteristics are presented as percentage for categorical data and mean value (standard deviation) for continuous variables. Differences between groups were analyzed by analysis of variance and Mantel Haenszel linear-by-linear trend as appropriate.

n = 64) at end of study. Marine n-3FA consumption remained stable during follow-up in the control group with a median increase of plasma PL marine n-3FA level of 0.1 wt% (IQR -0.8 to 1.0) and change in FFQ based marine n-3FA consumption estimates of -1.0 g/month (IQR -9.4 to 6.3).

At baseline, moderate correlations were found between FFQ based marine n-3FA consumption estimates and the reference biomarker plasma PL marine n-3FA levels (approach 1, $r_s = 0.43$, p<0.001, n = 132, Fig 2). A reasonable degree of agreement between the study FFQ estimates and the reference biomarker was shown in a Bland Altman plot (Fig 3) and significant bias was ruled out by a one-sample t-test (t = 0.04, p = 0.96). Two groups of outlier observations were identified. One group consisted of patients reporting high intake of marine n-3FAs but had average plasma PL marine n-3 FA levels. Another group with high or very high plasma PL marine n-3FA levels had average marine n-3FA consumption according to the study FFQ.

Correlations between FFQ based marine n-3FA consumption estimates and plasma PL marine n-3FA levels were stronger at the end of study ($r_s = 0.60$, p<0.001, n = 64, Fig 4) than at baseline. One-sample t-test (t = 0.06, p = 0.95) and a Bland Altman plot confirmed an acceptable degree of agreement between the two methods at this time point (Fig 5).

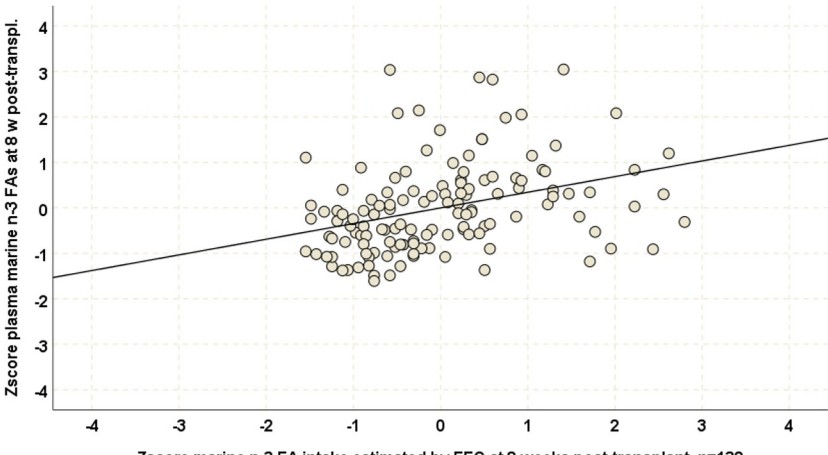

**Fig 2. Scatterplots of standardized plasma PL marine n-3FA levels and standardized FFQ based marine n-3FA consumption estimates with regression line at eight weeks post-transplant (n = 132).**

Baseline correlation analysis was repeated for patients belonging to the ORENTRA trial control group ($r_s$ = 0.45, p<0.001, n = 66, S5 Fig) and we found a high degree of agreement between the methods (S6 Fig), similar to what was shown for the whole study cohort at baseline.

We performed a multivariate stepwise forward regression analysis, adjusting for the potential confounding factors recipient age, gender, height, weight, body mass index, renal function, physical activity, educational level and smoking habits (p<0.10 for inclusion of variables in the final regression model) at baseline and end of study. The reference biomarker plasma PL marine n-3FA level was associated with FFQ based marine n-3FA consumption estimates (Std. β-coeff. 0.24, p = 0.01), as well as recipient age (Std. β-coeff. 0.25, p = 0.01) and smoking habits (Std. β-coeff. 0.15, p = 0.06) at baseline (n = 132). Together the three variables included in final

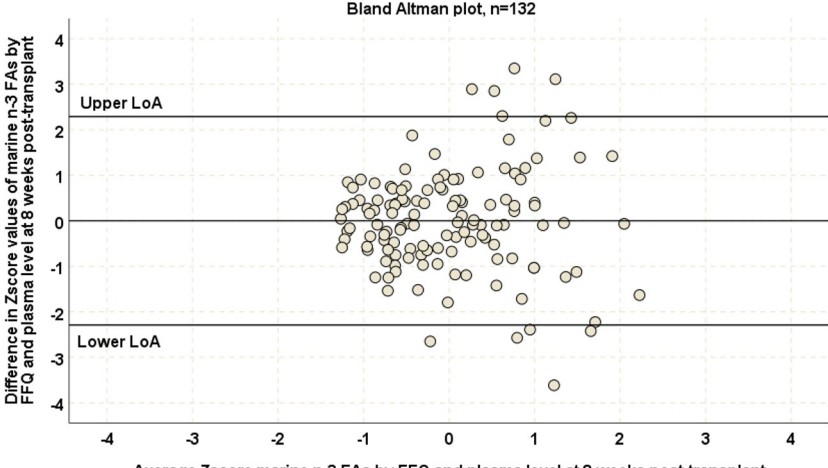

**Fig 3. Bland-Altman plot assessing degree of agreement between standardized plasma PL marine n-3FA levels and standardized FFQ based marine n-3FA consumption estimates at baseline eight weeks post-transplant.** We used standardization of data obtained by the study FFQ and reference biomarker, hence the mean value was set at 0. The upper and lower limits of agreement were set at 2 standard deviations from the mean. The Bland Altman plot includes all patients enrolled in the ORENTRA trial (n = 132) at the baseline time-point.

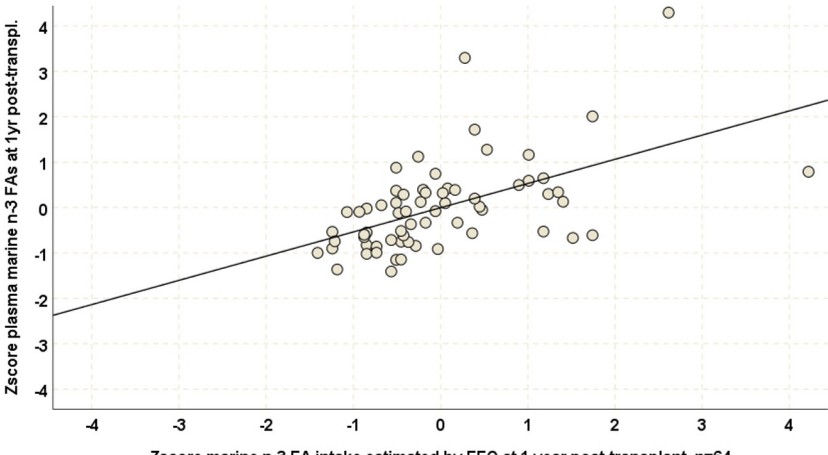

**Fig 4. Scatterplots of standardized plasma PL marine n-3FA levels and standardized FFQ based marine n-3FA consumption estimates with regression line at one year post-transplant for patients belonging to the control group of the ORENTRA trial (n = 64).**

regression model explained 23% of the variance in plasma PL marine n-3FA levels. At the end of study, only FFQ based marine n-3FA consumption estimates (Std. β-coeff. 0.54, p<0.001) was included in the final regression model, and it explained 29% of the variance in the reference biomarker.

Correlations between FFQ based marine n-3FA consumption estimates and the reference biomarker were slightly weaker for fatty fish intake (approach 2, baseline $r_s$ = 0.35 and end of study $r_s$ = 0.46) and number of fish servings per month (approach 3, baseline $r_s$ = 0.38 and end of study $r_s$ = 0.43) than for total marine n-3FA consumption estimates (approach 1). Correlations with the reference biomarker for individual food items included in the study FFQ are shown in S7 Fig. The food item cod liver oil showed a low correlation with the reference

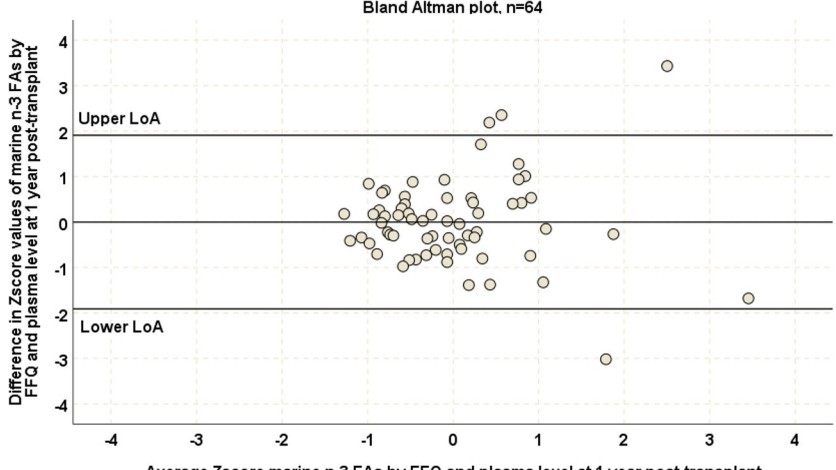

**Fig 5. Bland-Altman plot assessing degree of agreement between standardized plasma PL marine n-3FA levels and standardized FFQ based marine n-3FA consumption estimates at one year post-transplant.** We used standardization of data obtained by the study FFQ and reference biomarker, hence the mean value was set at 0. The upper and lower limits of agreement were set at 2 standard deviations from the mean. The Bland Altman plot includes only patients belonging to the control group of the ORENTRA trial (n = 64) at the end of study time-point.

biomarker at baseline ($r_s$ = 0.21). Marine n-3FA supplementation, including cod liver oil, was discontinued after enrollment in the ORENTRA trial and consequently intake of cod liver oil did not influence results at one year post-transplant.

## Discussion

The main finding of the present study was that marine n-3FA consumption estimates based on a new FFQ focused on fish consumption showed a moderate correlation with the reference biomarker plasma PL marine n-3FA levels at eight weeks post-transplant and a slightly stronger correlation at one year post-transplant. The correlations found in the present study are in the range of what is regarded as acceptable in FFQ validation studies [22].

Previous FFQs have mainly focused on fatty fish intake, assumed to reflect marine n-3FA consumption better than total fish intake [23–35]. We hypothesized that a more meticulous approach using weighted response scales based on EPA and DHA content in fatty and lean fish, other seafoods and marine n-3FA supplements would provide a more precise estimation of marine n-3FA consumption. In our cohort, approach 1, which estimated total marine n-3FA consumption from all the data obtained by the study FFQ, showed a stronger correlation with the reference biomarker than approach 2 (which only focused on fatty fish intake) and 3 (which used the number of fish servings), suggesting that our hypothesis was correct.

However, the study FFQ only provided slightly stronger correlations than most recent FFQs focused on fish consumption (Table 2) and the utility of the study FFQ will have to be confirmed by other studies before it can be used in epidemiological research.

Plasma PL marine n-3FA levels did not differ between baseline and end of study visits for the majority of patients. This is consistent with previous reports from large Norwegian cohorts and supports the notion that a single fatty acid measurement may be acceptable for epidemiological studies [3, 19]. However, the association between FFQ based marine n-3FA consumption estimates and the reference biomarker was stronger at end of study than at baseline. There could be several explanations to this finding. Study participants might have become more aware of their eating habits due to participation in the ORENTRAL trial and reported fish consumption more accurately when they completed the FFQ the second time. We found a

**Table 2. Summary of selected food frequency questionnaire validation studies published during the last six years, focusing on fish and/or marine fatty acid consumption, using circulating phospholipids or erytrocytes as the reference biomarker.**

| First author (reference) | Published, year | n | Study population | Reference marine fatty acid biomarker | Correlation coefficient |
|---|---|---|---|---|---|
| Giovannelli J [23] | 2014 | 2630 | General population | Plasma phospholipid | r = 0.39–0.43 |
| Lassale C [24] | 2016 | 198 | General population | Plasma phospholipid | $r_s$ = 0.51–0.54 |
| Sluik D [25] | 2016 | 383 | General population | Plasma phospholipid | r = 0.43–0.47 |
| Whitton C [26] | 2017 | 161 | General population | Plasma phospholipid | r = 0.36 |
| Laursen UB [27] | 2018 | 200 | General population | Plasma phospholipid | $r_s$ = 0.45 |
| Shen W [28] | 2019 | 108 | General population | Whole blood phospholipid | r = 0.67 |
| Schumacher TL [29] | 2016 | 39 | Hyperlipidemia | Erythrocyte | $r_s$ = 0.53–0.62 |
| Allaire J [30] | 2015 | 60 | Prostate cancer | Erythrocyte | $r_s$ = 0.59 |
| Brunvoll SH [31] | 2018 | 49 | Breast cancer | Serum phospholipid | r = 0.36–0.53 |
| Lepsch J [32] | 2014 | 248 | Pregnant women | Serum phospholipid | $r_s$ = 0.21–0.26 |
| Zhou YB [33] | 2017 | 804 | Pregnant women | Plasma phospholipid | $r_s$ = 0.35 |
| | | | | Erythrocyte | $r_s$ = 0.33 |
| Kobayashi M [34] | 2017 | 188 | Pregnant women | Serum phospholipid | $r_s$ = 0.33–0.45 |
| Liu MJ [35] | 2016 | 408 | Lactating women | Plasma phospholipid | $r_s$ = 0.36 |
| | | | | Erythrocyte | $r_s$ = 0.24 |

lower correlation with the reference biomarker for cod liver oil than for other food items in the study FFQ at baseline, which likely influenced the results. Some patients with high plasma PL marine n-3FA levels reported only average marine n-3FA intake according to the study FFQ, all of whom reported frequent use of cod liver oil. This signals that the study FFQ weighted response scale for cod liver oil likely underestimated marine n-3FA content, thus the study FFQ in its current form lacks precision for patients taking daily marine n-3FA supplements. Additionally, patients with average plasma PL marine n-3 FA levels who reported high levels of marine n-3FA intake in the FFQ, showed this pattern both at baseline and end of study. This might be due to social desirability bias and has likely influenced results at both time-points.

Fish intake in Norway is higher than in most other European countries, due to the rich fishing grounds along the Norwegian coastline with easy access to fresh cold-water fish [36]. Plasma PL marine n-3FA levels in the present cohort were relatively high, even for a Norwegian population, signaling a selected population that focuses on healthy eating habits. On the other hand, plasma PL marine n-3FA levels in the present study were comparable to a previous large cohort study in Norwegian RTRs, suggesting that the sample was representative of a Norwegian transplant cohort [3]. Confounding factors like socioeconomical class, educational level, smoking habits and physical activity may influence associations between fish intake and outcomes in epidemiological research [36]. In this cohort, FFQ based marine n-3FA consumption estimates and plasma PL marine n-3FA level were associated with smoking habits, but not other life-style factors.

Dietary habits are changing in the Nordic countries, with lower fish consumption in younger patients, including Norwegian RTRs [3], thus necessitating revision of questions and response categories for the present study FFQ in future studies. Cod liver oil intake is an old tradition in Norway [37] and was therefore included as one of the food times in the study FFQ. This may be omitted in areas where intake of cod liver oil or other marine n-3FA supplements are uncommon.

Strengths of the present study include a well-described cohort, plasma PL fatty acid analysis and a study FFQ performed at two time points, which might improve accuracy. The study FFQ has a simple design, is easy to read and understand and only takes a few minutes to answer, which is desirable in large epidemiological studies.

There were also several limitations, including limitations by design such as recall bias and social desirability bias and a relatively small sample size. The study FFQ marine n-3FA consumption estimates were based on the sum of weighted response scales for ten food items, containing questions on how frequent the food items were consumed, but not on portion size. Thus, the weighted responses used to calculate marine n-3FA intake were based on assumptions of standard portion size for each item, constituting a major limitation in the present study. Moreover, the study FFQ did not contain any questions regarding seasonal variations, which could be relevant for some of the included food items in a Norwegian cohort. The study FFQ contains rather detailed questions about fish and seafood intake and response categories with minor differences (Fig 1). This likely improved precision for patients who are well aware of their eating habits but could have been challenging for other patients, possibly leading to random responses. Broader response categories might have produced more reliable data [38]. For patients on marine n-3FA supplements, like cod liver oil, weighted responses for this food item in the study FFQ likely underestimated supplements as a source of marine n-3FAs.

The questionnaire was designed to estimate marine n-3FA consumption in a Norwegian transplant cohort. Due to dietary differences between regions and between patient populations, FFQ validation studies designed for one region or one particular target population may not apply to other regions or other patient cohorts [22]. In other regions, food items and weighted response scales should be revised to reflect fish consumption in that region.

Moreover, adjustment for portion size and seasonal variations can be made to improve FFQ performance.

In conclusion, marine n-3FA consumption estimates based on our study FFQ showed a moderate correlation with the reference biomarker plasma PL marine n-3FA levels, with comparable performance to previous FFQs. We recommend using fatty acid analysis to ensure objective measurement of marine n-3FA consumption in clinical trials, but our FFQ might be useful in epidemiological studies where resources are limited.

## Supporting information

**S1 Fig. Study food frequency questionnaire focusing on food items containing marine n-3 fatty acids (Norwegian version).** The study subjects responded to the question "During a typical month, how often do you eat these food items?" using one of six response alternatives for each of the ten food item categories.
(PDF)

**S2 Fig. Study food frequency questionnaire focusing on food items containing marine n-3 fatty acids (investigator's scoring sheet version in English).** The study subjects responded to the question "During a typical month, how often do you eat these food items?" using one of six response alternatives for each of the ten food item categories. Based on EPA and DHA content in the meat of various fish and other seafoods presented in the US Department of Agriculture Food Composition Database and assuming a standard portion size for dinner and bread spread, every potential response was given a weight (shown inside boxes). Total intake of marine n-3 fatty acids per month was calculated as the sum of the ten weighted responses in grams.
(PDF)

**S3 Fig. Study food frequency questionnaire focusing on food items containing marine n-3 fatty acids (investigator's scoring sheet version in English comprising fatty fish items only).** The study subjects responded to the question "During a typical month, how often do you eat these food items?" using one out of six response alternatives for each food item categories. Based on EPA and DHA content in the meat of various fish and other seafoods presented in the US Department of Agriculture Food Composition Database and assuming a standard portion size for dinner and bread spread, every potential response was given a weight (shown inside boxes). Total intake of marine n-3 fatty acids per month was calculated as the sum of the weighted responses in grams, which for fatty fish intake comprised the four items shown.
(PDF)

**S4 Fig. Study food frequency questionnaire focusing on food items containing marine n-3 fatty acids (investigator's scoring sheet version in English comprising fish servings per month).** The study subjects responded to the question "During a typical month, how often do you eat these food items?" using one out of six response alternatives for each of the ten food item categories. Servings of fish per month was calculated as the sum of the ten responses, using the center value for each response category as shown.
(PDF)

**S5 Fig. Scatterplots of standardized plasma marine n-3FA levels and standardized FFQ based marine n-3FA consumption estimates with regression lines at baseline eight weeks post-transplant for patients belonging to the control group of the ORENTRA trial (n = 66).**
(TIF)

**S6 Fig. Bland-Altman plot assessing degree of agreement between standardized plasma marine n-3FA levels and standardized FFQ based marine n-3FA consumption estimates at baseline eight weeks post-transplant for patients belonging to the control group of the ORENTRA trial (n = 66).**
(TIF)

**S7 Fig. Correlation matrix presenting Spearman's correlation coefficients at eight weeks (baseline visit) after renal transplantation for the whole study population (n = 132) and one year post-transplant (end of study visit) for patients belonging to the control group of the ORENTRA trial (n = 64).**
(TIF)

**S1 File. Supporting information.** Includes information regarding "Patient screening and recruitment in the ORENTRA trial", "Fatty acid analysis", "Sample Size and Power Calculation" and "Development of the study Food Frequency Questionnaire".
(DOCX)

## Acknowledgments

We thank coworkers Rikke Bülow Eschen, Annette Andreassen, Birthe H. Thomsen and Inge Aardestrup at The Lipid Research Laboratory, Aalborg University Hospital, Denmark for analyzing plasma phospholipid fatty acids. We thank statistician Owen Thomas and colleague dr. Anupam Chandra at Akershus University Hospital for their contribution to this manuscript. We thank the funding sources Gidske and Peter Jacob Sørensen Research Fund and the South-Eastern Norway Regional Health Authority. Finally, we thank the study participants in the ORENTRA trial.

## Author Contributions

**Conceptualization:** My Svensson, Trond Jenssen, Erik B. Schmidt, Ivar A. Eide.

**Data curation:** Ivar A. Eide.

**Formal analysis:** Joe Chan, Ivar A. Eide.

**Funding acquisition:** My Svensson.

**Investigation:** Erik B. Schmidt, Ivar A. Eide.

**Methodology:** Joe Chan, My Svensson, Trond Jenssen, Erik B. Schmidt, Ivar A. Eide.

**Project administration:** Trond Jenssen, Ivar A. Eide.

**Resources:** My Svensson, Trond Jenssen, Erik B. Schmidt, Ivar A. Eide.

**Supervision:** My Svensson, Erik B. Schmidt, Ivar A. Eide.

**Validation:** Ivar A. Eide.

**Writing – original draft:** Joe Chan.

**Writing – review & editing:** Joe Chan, My Svensson, Trond Jenssen, Erik B. Schmidt, Ivar A. Eide.

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
