## [Decision Letter · Decision Letter 0]

8 Oct 2020

PONE-D-20-12551

Marine fatty acid consumption in a Norwegian renal transplant cohort: comparison of a food frequency questionnaire with plasma n-3 levels

PLOS ONE

Dear Dr. Chan,

Thank you for submitting your manuscript to PLOS ONE. After careful consideration, we feel that it has merit but does not fully meet PLOS ONE’s publication criteria as it currently stands. Therefore, we invite you to submit a revised version of the manuscript that addresses the points raised during the review process.

The manuscript has been evaluated by three reviewers, and their comments are available below.

The reviewers have raised a number of concerns that need attention. They request additional information on methodological aspects of the study, and revisions to the statistical analyses. 

Could you please revise the manuscript to carefully address the concerns raised?

We look forward to receiving your revised manuscript.

Kind regards,

Carmen Melatti

Associate Editor

PLOS ONE

Journal Requirements:

3. Please provide additional details regarding participant consent. In the ethics statement in the Methods and online submission information, please ensure that you have specified (1) whether consent was informed and (2) what type you obtained (for instance, written or verbal). If your study included minors, state whether you obtained consent from parents or guardians. If the need for consent was waived by the ethics committee, please include this information.

4. Please note that according to our submission guidelines (http://journals.plos.org/plosone/s/submission-guidelines), outmoded terms and potentially stigmatizing labels should be changed to more current, acceptable terminology. For example: “Caucasian” should be changed to “white” or “of [Western] European descent” (as appropriate).

5. In your Methods section, please provide additional information about the related clinical trial (fir example please describe how plasma n-3 levels were measured, and what was the intervention) and about the  participant recruitment method and the demographic details of your participants. Please ensure you have provided sufficient details to replicate the analyses such as: a) the recruitment date range (month and year), b) a description of any inclusion/exclusion criteria that were applied to participant recruitment, c) a table of relevant demographic details, d) a statement as to whether your sample can be considered representative of a larger population, e) a description of how participants were recruited, and f) descriptions of where participants were recruited and where the research took place.

Reviewers' comments:

Reviewer's Responses to Questions

**Comments to the Author**

1. Is the manuscript technically sound, and do the data support the conclusions?

Reviewer #1: Yes

Reviewer #2: Partly

Reviewer #3: Partly

2. Has the statistical analysis been performed appropriately and rigorously? 

Reviewer #1: Yes

Reviewer #2: No

Reviewer #3: No

3. Have the authors made all data underlying the findings in their manuscript fully available?

Reviewer #1: Yes

Reviewer #2: Yes

Reviewer #3: Yes

4. Is the manuscript presented in an intelligible fashion and written in standard English?

Reviewer #1: Yes

Reviewer #2: Yes

Reviewer #3: Yes

5. Review Comments to the Author

Reviewer #1: The authors validate a new and simple food frequency questionnaire to determine total PUFAn3 level on patients, in their case with kidney transplant.

The text is interesting but there are some not clear points.

In the introduction, line 53-54, they define EPA and DHA as "essential marine n-3FAs", I understand what they mean, but I suggest to modify the sentence because the essential n3 FA is ALA.

On line 55, authors describe fatty acid analysis as "expensive". Fatty acid analysis with GC is not expensive, it is better if they can report data and cost about it.

Regarding the rs difference between first test and end of study, authors can exclude that the difference is due by the different sample size (132 vs 71), it could be good to show the rs at the start of study regarding the 71 patients present at the end of the study.

The weak point of this study is, as the authors themselves said, is that the presented FFQ is strongly linked to Norwegian diet.

Reviewer #2: Given the limited data analysis, the authors claim to have shown that n-3FA consumption estimates based on the FFQ demonstrated a moderate correlation with the reference biomarker plasma phospholipid n-3FA levels. The correlation and trend test were the primary analyses attempted. The trend test was certainly of interest. However, there are some concerns.

1. This is basically a convenience sample of 132 subjects. The investigators should provide a statistical rationale for this sample size and its adequacy from a power perspective.

2. Given the extensive patient characteristic data in Table 1, why wasn’t a multivariate analysis attempted to show the effect of adjustment of the variables in a reasonable format?

Reviewer #3: This manuscript reports on the evaluation of a food frequency questionnaire to assess n-3FA consumption. It is a manuscript of scientific relevance, but it has inadequate statistical analysis for the purpose of the study. The authors should use agreement methods to complementing statistical analyzes.

Moreover, “Methods and reagents” are not described in sufficient detail for another researcher to reproduce the experiments described.

Other specific comments:

1. Introduction: Renal function benefits are reported only by n-3 FA supplementation? And the objective of this study was to evaluate a new FFQ to estimates n-3 FA consumption. The arguments do not make sense due to inconsistency.

2. Line 63: “Methods and reagents” are not described in sufficient detail for another researcher to reproduce the experiments described.

3. Line 73: Please explain fatty acid extraction method.

4. Lines 92-93: Standard portion size for dinner and bread spread was estimated after the study subjects responded to the FFQ? Portions size are not in the questionnaire.

5. Line 98: Correlation analysis are not appropriate to asses the agreement of two methods. Use agreement methods to complementing statistical analyzes. Bland-Altman is a reliable approach for statistical analysis.

6. Lines 99-102: This is “study participants”. Remove from “statistical analysis”.

7. Line 108: Results and Discussion should be reviewed after further statistical analysis

6. PLOS authors have the option to publish the peer review history of their article (what does this mean?). If published, this will include your full peer review and any attached files.

Reviewer #1: No

Reviewer #2: No

Reviewer #3: No

---

## [Author Response · Author response to Decision Letter 0]

23 Nov 2020

Thank you for the opportunity to submit a revised version of this manuscript. The Reviewers provided many insightful comments, for which we are very grateful. We have revised the manuscript and figures according to these comments and performed additional statistical analyses like multivariate regression and Bland Altman plots. We also include a Supporting Information File presenting patient recruitment, fatty acid analysis and the study FFQ in detail as requested. 

The revised manuscript contains two tables and five figures. We have included Bland Altman plots in the revised manuscript, which also marks a shift in statistical approach, as suggested by Reviewer 2 and Reviewer 3. We feel that these changes, which are highlighted in the track changes version, have improved the quality of the manuscript, and that study limitations and key information have been made clearer and more available to the reader in the revised manuscript. 

Point to point responses:

Editor: 

Reply: We have made our best effort to ensure that the manuscript complies with PLOS ONE’s style requirements. Formatting of the tables has been reworked. Please let us know if there are still requirements we have failed to notice. 

Reply: Please find a copy of the questionnaire in English and Norwegian and a scoring sheet as supporting figures. The last Author developed the questionnaire in Norwegian and in English and tested it in a pilot study (n=10 Norwegian RTRs) before it was implemented in the ORENTRA trial. Together with a professional translator he performed translations back and forth to ensure adequacy of the English version, thus both the Norwegian and English versions should be considered the original questionnaire. There is no copyright. 

In the revised manuscript we discuss potential limitations of the present study FFQ, including lack of questions on portion size and seasonal variation, and highlight that questions and response categories should be revised to reflect fish consumption in the region and/or the patient population at hand. This should enable other researchers not only to replicate, but also to adequately revise the FFQ to improve performance in future research. 

3. Please provide additional details regarding participant consent. In the ethics statement in the Methods and online submission information, please ensure that you have specified (1) whether consent was informed and (2) what type you obtained (for instance, written or verbal). If your study included minors, state whether you obtained consent from parents or guardians. If the need for consent was waived by the ethics committee, please include this information.

Reply: Please find a more detailed description of the written informed consent needed to participate in the study in the revised manuscript. Only adult RTRs participated in the ORENTRA trial, which is highlighted in the revised manuscript. 

4. Please note that according to our submission guidelines, outmoded terms and potentially stigmatizing labels should be changed to more current, acceptable terminology. For example: “Caucasian” should be changed to “white” or “of [Western] European descent” (as appropriate).

Reply: We apologize for this. The word “Caucasian” is changed to “White”.

5. In your Methods section, please provide additional information about the related clinical trial (for example please describe how plasma n-3 levels were measured, and what was the intervention) and about the participant recruitment method and the demographic details of your participants. Please ensure you have provided sufficient details to replicate the analyses such as: a) the recruitment date range (month and year), b) a description of any inclusion/exclusion criteria that were applied to participant recruitment, c) a table of relevant demographic details, d) a statement as to whether your sample can be considered representative of a larger population, e) a description of how participants were recruited, and f) descriptions of where participants were recruited and where the research took place.

Reply: Certainly. The Method section has been revised according to comments put forward by Editor and Reviewers and we included a Supporting Information File as recommended by the Editor. We chose to put the majority of additional information on methods (a, b and e) in the Supporting Information File, as the Method section would otherwise be too long and hard to follow. However, we refer to what information can be found in the Supporting Information (including a, b and e) in the revised manuscript. The manuscript already includes Table 1 with demographic data (c). Description of where participants were recruited and where the research took place are now mentioned in the revised manuscript and further explored in the Supporting Information File (f). The revised manuscript now contains clear statements on whether the sample is representative of a larger population (d) and highlights the need for confirmatory studies. 

Reviewer #1: 

The authors validate a new and simple food frequency questionnaire to determine total PUFAn3 level on patients, in their case with kidney transplant. The text is interesting but there are some not clear points.

In the introduction, line 53-54, they define EPA and DHA as "essential marine n-3FAs", I understand what they mean, but I suggest to modify the sentence because the essential n3 FA is ALA.

Reply: Thank you for pointing this out. We have replaced the phrase “n-3FA” with “marine n-3FA” throughout the revised manuscript to avoid confusing the reader. This also includes the study title. 

On line 55, authors describe fatty acid analysis as "expensive". Fatty acid analysis with GC is not expensive, it is better if they can report data and cost about it.

Reply: We apologize for this rather bombastic statement, which has been rephrased in the revised manuscript. GC was performed at a research laboratory in Aarhus, Denmark. We didn’t feel comfortable including data on GC costs in the manuscript. Laboratory costs partly depend on coworker salary and differ considerably between regions, thus adding this information could potentially be misleading.

Regarding the rs difference between first test and end of study, authors can exclude that the difference is due by the different sample size (132 vs 71), it could be good to show the rs at the start of study regarding the 71 patients present at the end of the study.

Reply: We agree. In the revised manuscript, we include data for patients in the control group of the ORENTRA trial at baseline 8 weeks (n=66) and end of study 1 year post-transplant (n=64, two patients did not meet at the end of study visit). 

In the original manuscript, patients who withdrew early from the intervention group were added to the control group at the end of study visit, i.e., included in statistical analyses. This might confuse the reader, particularly when we include correlation analysis for patients belonging to the control group at baseline, as requested by Reviewer 1. Thus, we chose to exclude all patients randomized to the intervention group of the ORENTRA trial for statistical analysis both at the end of study and in the baseline subgroup analysis that Reviewer 1 requested. 

At baseline, we found statistically significant and very similar correlations between the study FFQ estimates and the reference biomarker in the control group (n=66) and the whole study cohort (n=132), suggesting that sample size did not influence results to a major extent.

The weak point of this study is, as the authors themselves said, is that the presented FFQ is strongly linked to Norwegian diet.

Reply: You are quite right. That said, we speculate (not included in the manuscript) that with revision of questions and response categories to reflect regional fish intake, this approach may work better in populations with low intake of marine n-3FA supplements. We have included a paragraph in the revised manuscript discussing reasons for the lower correlation between the study FFQ estimates and the reference biomarker at baseline, where use of cod liver oil at the baseline time-point could be one reason, as illustrated under (not in manuscript): (Figure in the Word-document).

Reviewer #2: 

Given the limited data analysis, the authors claim to have shown that n-3FA consumption estimates based on the FFQ demonstrated a moderate correlation with the reference biomarker plasma phospholipid n-3FA levels. The correlation and trend test were the primary analyses attempted. The trend test was certainly of interest. However, there are some concerns. 

1. This is basically a convenience sample of 132 subjects. The investigators should provide a statistical rationale for this sample size and its adequacy from a power perspective.

Reply: Thank you for pointing this out. The present study cohort consisted of patients who were enrolled in the ORENTRA trial (n=132). Power calculation for this trial was based on the primary endpoint renal function, which for the study FFQ makes this is a convenience sample – you are absolutely right. That said, the clinical steering committee discussed power perspectives upfront also for secondary endpoints, including the study FFQ. Comparable studies used power estimations based on a correlation coefficient of at least 0.3 (95% CI and 20% drop-out rate was used for calculating sample size in these studies), necessitating a sample size of ≥ 84 patients. For r ≥ 0.4 we need ≥ 46 patients, for r ≥ 0.5 we need ≥ 29 patients and for r ≥ 0.6 we need ≥ 19 patients. Based on these reports it seemed obvious that the sample size in the present study was adequate from a power perspective. We have included this information in the Supporting Information file.

2. Given the extensive patient characteristic data in Table 1, why wasn’t a multivariate analysis attempted to show the effect of adjustment of the variables in a reasonable format?

Reply: We agree that a multivariate regression analysis in addition to the univariate (correlation) analysis was of interest, and it has been included in the revised manuscript.

Reviewer #3: 

This manuscript reports on the evaluation of a food frequency questionnaire to assess n-3FA consumption. It is a manuscript of scientific relevance, but it has inadequate statistical analysis for the purpose of the study. The authors should use agreement methods to complementing statistical analyzes.

Reply: We agree. Please find Bland Altman plots where we use z-statistics to standardize both study FFQ estimates and the reference biomarker. This allows for a more meaningful comparison of methods.

Moreover, “Methods and reagents” are not described in sufficient detail for another researcher to reproduce the experiments described.

Reply: We apologize for this. Please find a detailed description of methods in the revised manuscript and particularly in the Supporting Information File.

Other specific comments:

1. Introduction: Renal function benefits are reported only by n-3 FA supplementation? And the objective of this study was to evaluate a new FFQ to estimates n-3 FA consumption. The arguments do not make sense due to inconsistency.

Reply: We have rephrased the Introduction to make the aim of this study clear to the reader. 

2. Line 63: “Methods and reagents” are not described in sufficient detail for another researcher to reproduce the experiments described.

Reply: A detailed description of methods can be found in the Supporting Information.

3. Line 73: Please explain fatty acid extraction method.

Reply: A detailed description of the fatty acid analysis including FA extraction can be found in the Supporting Information.

4. Lines 92-93: Standard portion size for dinner and bread spread was estimated after the study subjects responded to the FFQ? Portions size are not in the questionnaire.

Reply: Correct. This is highlighted as a major limitation in the revised manuscript. We performed an additional correlation analysis where FFQ based marine n-3FA consumption estimates were adjusted for patient weight (g/month/kg) and correlations with the reference biomarker were nearly identical to unadjusted estimates, signaling that lack of portion size estimates likely did not strongly influence on results. Nonetheless, it is obvious that our results were hampered by lack of portion size estimates to some extent and should have been included in the FFQ. For the record, in the revised manuscript we also address that the study FFQ lacked questions on seasonal variation for the food items included. For most regions, seasonal variations would not constitute a major problem, but for some coastal regions in Norway this could influence results. Future versions of the FFQ should include portion size and seasonal variations, and this has been made clear to the reader in the revised manuscript.

5. Line 98: Correlation analysis are not appropriate to asses the agreement of two methods. Use agreement methods to complementing statistical analyzes. Bland-Altman is a reliable approach for statistical analysis.

Reply: Thank you for pointing this out. Please find Bland-Altman plots in Figure 3 and 5.

6. Lines 99-102: This is “study participants”. Remove from “statistical analysis”.

Reply: We have rephrased this sentence.

7. Line 108: Results and Discussion should be reviewed after further statistical analysis

Reply: Certainly. Revision of the statistical approach produced new data, which we present in the Result section and we address this data in the Discussion section.

---

## [Editor Report · Decision Letter 1]

3 Dec 2020

Marine n-3 fatty acid consumption in a Norwegian renal transplant cohort: comparison of a food frequency questionnaire with plasma phospholipid marine n-3 levels

PONE-D-20-12551R1

Dear Dr. Chan,

We’re pleased to inform you that your manuscript has been judged scientifically suitable for publication and will be formally accepted for publication once it meets all outstanding technical requirements.

Kind regards,

Stefano Turolo, BD

Guest Editor

PLOS ONE
---

## [Editor Report · Acceptance letter]

9 Dec 2020

PONE-D-20-12551R1 

Marine n-3 fatty acid consumption in a Norwegian renal transplant cohort: comparison of a food frequency questionnaire with plasma phospholipid marine n-3 levels 

Dear Dr. Chan:

I'm pleased to inform you that your manuscript has been deemed suitable for publication in PLOS ONE. Congratulations! Your manuscript is now with our production department. 

Kind regards, 

on behalf of

Dr. Stefano Turolo 

Guest Editor

PLOS ONE